WIN 55,212-2 shows anti-inflammatory and survival properties in human iPSC-derived cardiomyocytes infected with SARS-CoV-2

http://orcid.org/0000-0003-4514-5353 Aragão Luiz Guilherme H. S. 1
Oliveira Júlia T. 1
http://orcid.org/0000-0002-8092-2149 Temerozo Jairo R. 2 3
Mendes Mayara A. 1
http://orcid.org/0000-0003-0511-7083 Salerno José Alexandre 4
Pedrosa Carolina S. G. 1
Puig-Pijuan Teresa 1 5
Veríssimo Carla P. 4
Ornelas Isis M. 1
Torquato Thayana 1
http://orcid.org/0000-0002-2636-9186 Vitória Gabriela 1
Sacramento Carolina Q. 3 6
Fintelman-Rodrigues Natalia 3 6
da Silva Gomes Dias Suelen 6
http://orcid.org/0000-0002-0552-5995 Cardoso Soares Vinicius 6 7
Souza Letícia R. Q. 1
http://orcid.org/0000-0003-4383-8746 Karmirian Karina 1 4
Goto-Silva Livia 1
http://orcid.org/0000-0002-5872-6133 Biagi Diogo 8
Cruvinel Estela M. 8
http://orcid.org/0000-0003-0957-1259 Dariolli Rafael 8 9
http://orcid.org/0000-0001-7072-7879 Furtado Daniel R. 1
Bozza Patrícia T. 6
Borges Helena L. 4
Souza Thiago M. L. 3 6
Guimarães Marília Zaluar P. 1 4
Rehen Stevens K. 1 10 srehen@lance-ufrj.org
1 D’Or Institute for Research and Education (IDOR) , Rio de Janeiro, Rio de Janeiro , Brazil
2 Laboratory on Thymus Research, Oswaldo Cruz Institute (IOC) , Rio de Janeiro, Rio de Janeiro , Brazil
3 National Institute for Science and Technology on Innovation in Diseases of Neglected Populations (INCT/IDPN), Center for Technological Development in Health (CDTS), Oswaldo Cruz Foundation (Fiocruz) , Rio de Janeiro, Rio de Janeiro , Brazil
4 Institute of Biomedical Sciences, Federal University of Rio de Janeiro (UFRJ) , Rio de Janeiro, Rio de Janeiro , Brazil
5 Carlos Chagas Filho Institute of Biophysics, Federal University of Rio de Janeiro (UFRJ) , Rio de Janeiro, Rio de Janeiro , Brazil
6 Laboratory of Immunopharmacology, Oswaldo Cruz Institute (IOC), Oswaldo Cruz Foundation (Fiocruz) , Rio de Janeiro, Rio de Janeiro , Brazil
7 Program of Immunology and Inflammation, Federal University of Rio de Janeiro (UFRJ) , Rio de Janeiro, Rio de Janeiro , Brazil
8 Pluricell Biotech , São Paulo, São Paulo , Brazil
9 Department of Pharmacological Sciences, Icahn School of Medicine at Mount Sinai , New York, New York , United States
10 Department of Genetics, Institute of Biology, Federal University of Rio de Janeiro (UFRJ) , Rio de Janeiro, Rio de Janeiro , Brazil
Connor Mark
Electronic publication date: 2021 Oct 8
Publication date: 2021
Volume: 9
Electronic Location ID: e12262
Received 2021 Jun 15; Accepted 2021 Sep 16
Copyright: © 2021 Aragão et al.
Copyright year: 2021
Copyright holder: Aragão et al.
License: This is an open access article distributed under the terms of the Creative Commons Attribution License, which permits unrestricted use, distribution, reproduction and adaptation in any medium and for any purpose provided that it is properly attributed. For attribution, the original author(s), title, publication source (PeerJ) and either DOI or URL of the article must be cited.
License URL: https://creativecommons.org/licenses/by/4.0/

Keywords: Cannabinoids; SARS-Cov-2; COVID-19; Human iPSC-derived cardiomyocytes; WIN 55,212-2

Funding: National Council of Scientific and Technological Development (CNPq) 313688/2020-6 D’Or Institute for Research and Education (IDOR) This work was supported by the National Council of Scientific and Technological Development (CNPq) (Grant Number: 313688/2020-6) and scholarship grants from Coordination for the Improvement of Higher Education Personnel (CAPES) and CNPq. Intramural grants were provided from the D’Or Institute for Research and Education (IDOR). The funders had no role in study design, data collection and analysis, decision to publish, or preparation of the manuscript.

==============================
Coronavirus disease 2019 (COVID-19) is caused by severe acute respiratory syndrome coronavirus 2 (SARS-CoV-2), which can infect several organs, especially impacting respiratory capacity. Among the extrapulmonary manifestations of COVID-19 is myocardial injury, which is associated with a high risk of mortality. Myocardial injury, caused directly or indirectly by SARS-CoV-2 infection, can be triggered by inflammatory processes that lead to damage to the heart tissue. Since one of the hallmarks of severe COVID-19 is the “cytokine storm”, strategies to control inflammation caused by SARS-CoV-2 infection have been considered. Cannabinoids are known to have anti-inflammatory properties by negatively modulating the release of pro-inflammatory cytokines. Herein, we investigated the effects of the cannabinoid agonist WIN 55,212-2 (WIN) in human iPSC-derived cardiomyocytes (hiPSC-CMs) infected with SARS-CoV-2. WIN did not modify angiotensin-converting enzyme II protein levels, nor reduced viral infection and replication in hiPSC-CMs. On the other hand, WIN reduced the levels of interleukins six, eight, 18 and tumor necrosis factor-alpha (TNF-α) released by infected cells, and attenuated cytotoxic damage measured by the release of lactate dehydrogenase (LDH). Our findings suggest that cannabinoids should be further explored as a complementary therapeutic tool for reducing inflammation in COVID-19 patients.

Introduction

The causative agent of Coronavirus disease 2019 (COVID-19), SARS-CoV-2, can affect multiple organs, including lungs, nervous system (Carod-Artal, 2020), digestive system (Chen et al., 2020b), urinary system (Puelles et al., 2020), skin (Mahé et al., 2020; Diaz-Guimaraens et al., 2020) and heart (Maisch, 2020; Varga et al., 2020; Zheng et al., 2020).

A post-mortem study of a child with COVID-19 revealed diffuse myocardial interstitial inflammation with immune cells infiltration and necrosis (Dolhnikoff, 2020). Recently, we showed cardiac damage, namely microthrombi in small arteries and focal mild lymphocytic infiltrate in the ventricles, of an infant who died of COVID-19 (Gomes et al., 2021). Another study detected SARS-CoV-2 in myocardial tissue, which expressed inflammatory mediators, such as tumor necrosis factor-alpha (TNF-α), interferon-gamma (IFN-γ), chemokine ligand 5, as well as interleukin (IL) −6, −8, and −18 (Lindner et al., 2020). Additionally, patients with COVID-19 presented elevated levels of creatine kinase and lactate dehydrogenase (LDH) activity, which are biomarkers of heart injury (Chen et al., 2020b; Zhou et al., 2020).

High expression of Angiotensin-Converting Enzyme II (ACE2) in the heart has been correlated with severe COVID-19 and susceptibility of patients with pre-existing cardiac conditions (Chen et al., 2020a; Thum, 2020; Sharma et al., 2020; Dariolli et al., 2021). In vitro studies have shown that SARS-CoV-2 infects iPSC-derived cardiomyocytes (hiPSC-CMs) through ACE2 (Sharma et al., 2020; Dariolli et al., 2021), leading to upregulation of inflammation-related genes, including IL-6, IL-8, and TNF-α (Wong et al., 2020; Kwon et al., 2020). The increase in proinflammatory cytokines can cause several adverse effects in cardiomyocytes including arrhythmia (Keck et al., 2019), cellular hypertrophy (Smeets et al., 2008), cell death (Wang et al., 2016), conversion of fibroblasts into myofibroblasts (Wang et al., 2016) and alteration of action potentials’ duration (Aromolaran et al., 2018). The correlation between inflammation and heart damage in post-mortem and in vitro studies points to the need for finding strategies that mitigate direct SARS-CoV-2 cardiac outcomes.

For many centuries Cannabis sp. has been used for medicinal purposes and, more recently, it has been investigated as a therapeutic agent for cardiovascular diseases (Mendizábal & Adler-Graschinsky, 2007; Pacher et al., 2018). Cannabis has several known compounds, named phytocannabinoids, including delta-9-tetrahydrocannabinol (THC), which is the most abundant and the main psychoactive ingredient, followed in amount by cannabidiol (CBD). Besides phytocannabinoids, there is intensive research on endocannabinoids, such as anandamide and 2-arachidonoylglycerol, and synthetic cannabinoids, such as WIN 55,212–2 (WIN). Nguyen et al. (2021) showed the potential of cannabinoids to decrease SARS-CoV-2 infection, viral replication, and inflammation that are directly related to COVID-19 severity. Treatment with Cannabis extracts decreased ACE2 expression in oral, intestinal, and airway epithelia in vitro (Wang et al., 2020). It is noteworthy that cannabinoids have anti-inflammatory properties and exert their biological effect mainly by interaction with the cannabinoid receptors type 1 (CB1) and/or type 2 (CB2), to both of which WIN has high affinity and efficacy (Devane et al., 1988; Munro, Thomas & Abu-Shaar, 1993; Felder et al., 1995; Soethoudt et al., 2017; Sachdev et al., 2019). For instance, WIN was shown to reduce the number of lipopolysaccharide-activated microglia in the brain of an animal model of chronic inflammation (Marchalant, Rosi & Wenk, 2007). Another work showed that WIN decreased TNF-α and IL-6 plasma levels and myeloperoxidase activity in mice with experimental colitis (Feng et al., 2016). An extract fraction from Cannabis sativa Arbel strain enriched in CBD, cannabigerol and tetrahydrocannabivarin presented anti-inflammatory activity in lung epithelial cells treated with TNF-α but another fraction with high CBD containing terpenes in addition to phytocannabinoids enhanced proinflammatory parameters of macrophages (Anil et al., 2021). Additionally, high-CBD Cannabis sativa extracts presented anti-inflammatory properties in the epithelia pretreated with TNF-α and IFN-γ (Lei et al., 2020). Smith, Terminelli & Denhardt (2000) showed that treatment with WIN decreased serum TNF-α and IL-12 and increased IL-10 through the CB1 receptor in mice treated with lipopolysaccharide (Smith, Terminelli & Denhardt, 2000). Investigating the anti-inflammatory potential of Cannabis sativa in cardiomyocytes is important because the “cytokine storm” is a hallmark of COVID-19 and the cardiovascular system is mostly affected in severe cases (Unudurthi et al., 2020). To date, the effects of cannabinoids in human cardiomyocytes infected with SARS-CoV-2 has not been addressed.

In this work, we aimed to investigate the effects of a synthetic cannabinoid, that acts as a mixed CB1/CB2 receptors agonist, in hiPSC-CMs infected by SARS-CoV-2. WIN presented anti-inflammatory and protective properties by reducing the levels of proinflammatory cytokines and cell death in hiPSC-CM but did neither modulate ACE2 nor reduced SARS-CoV-2 infection and replication. Our data suggest that the anti-inflammatory and protective properties of WIN may be used to control inflammation and tissue damage during SARS-CoV-2 infection of heart cells.

Materials & methods

Chemical

WIN 55,212-2 mesylate was purchased from TargetMol (T4458). Stock solutions were prepared using 100% dimethyl sulfoxide (DMSO; D2650-Sigma-Aldrich, St. Louis, MO, USA) and sterile-filtered. The final concentration of DMSO in work solution was 0.01%.

iPS-cardiomyocyte differentiation and purification

hiPSC-CMs were purchased from Pluricell (São Paulo, Brazil) and used between day 25 and day 35 of differentiation. The hiPSC-CMs used here were generated and previously characterized in vitro by Cruvinel et al. (2020). Briefly, the enrichment of the cardiomyocyte population was assessed by flow cytometry and immunofluorescence of TNNT2, a specific marker, which revealed that, on average, 88.4% (+/− 8.4%) of cells were positive cells (Fig. S4). hiPSC-CMs were handled in four different groups: MOCK and SARS-CoV-2 (SARS-CoV-2 infection without WIN), which were also analyzed as controls in Salerno et al. (2021), MOCK WIN (no SARS-CoV-2 infection + WIN), and SARS-CoV-2 WIN (SARS-Cov-2 infection + WIN). All WIN-treated hiPSC-CMs were pretreated for 24 h with one µM WIN. Fresh culture medium with (or without) one µM WIN, combined or not with SARS-CoV-2, was added for 24 h to each experimental group, respectively.

SARS-CoV-2 propagation

SARS-CoV-2 was expanded in Vero E6 cells from an isolate of a nasopharyngeal swab obtained from a confirmed case in Rio de Janeiro, Brazil (GenBank accession no. MT710714). Viral isolation was performed after a single passage in 150 cm2 flasks cultured with high glucose DMEM plus 2% FBS. Observations for cytopathic effects were performed daily and peaked 4 to 5 days after infection. All procedures related to virus culture were handled in biosafety level 3 (BSL3) multi-user facilities according to WHO guidelines. Virus titers were determined as plaque-forming units (PFU/mL) as explained below, and virus stocks were kept at −80 °C.

SARS-CoV-2 titration

For virus titration, monolayers of Vero E6 cells (2 × 104 cell/well) in 96-well plates were infected with serial dilutions of supernatants containing SARS-CoV-2 for 1 h at 37 °C. A semi-solid high glucose DMEM medium containing 2% FSB and 2.4% carboxymethylcellulose was added and cultures were incubated for 3 days at 37 °C. Then, the cells were fixed with 10% formalin for 2 h at room temperature. The cell monolayer was stained with 0.04% solution of crystal violet in 20% ethanol for 1 h. Plaque numbers were scored in at least three replicates per dilution by independent readers blinded to the experimental group and the virus titers were determined by plaque-forming units (PFU) per milliliter.

SARS-CoV-2 infection

hiPSC-CMs were infected with SARS-CoV-2 at a multiplicity of infection (MOI) of 0.1 in high glucose DMEM without serum. After 1 h, cells were washed and incubated with Complete Medium with or without treatments for 48–72 h. Next, the supernatant was collected and cells were fixed with 4% paraformaldehyde (PFA) solution for posterior analysis.

Measurement of cytokines mediators and LDH cytotoxicity

Cytokines (IL-6, IL-7, IL-8, and TNF-α) were quantified in the supernatants from hiPSC-CMs samples by ELISA (R&D Systems, Minneapolis, MN, USA) following manufacturer’s instructions. The analysis of data was performed using software provided by the manufacturer (Bio-Rad Laboratories, Hercules, CA, USA). A range of 0.51–8,000 pg/mL recombinant cytokines was used to establish standard curves and the sensitivity of the assay. Cell death was determined according to the activity of lactate dehydrogenase (LDH) in the culture supernatants using a CytoTox® Kit (Promega, Madison, WI, USA) according to the manufacturer’s instructions.

Gene expression analysis

Qualitative endpoint PCR reactions were executed with the following primer sequences: CB1 (forward 5′-ATGTGGACCATAGCCATTGTG-3′; reverse: 5′-CCGATCCAGAACATCAGGTAGG-3′) and CB2 (forward 5′-GCTATCCACCTTCCTACAAAGC-3′; reverse: 5′-CTCAGCAGGTAGTCATTGGGG-3′). Glyceraldehyde-3-phosphate Dehydrogenase (GAPDH; forward: 5′-TTCGACAGTCAGCCGCATC-3′; reverse: 5′-GACTCCACGACGTACTCAGC-3′) was used as the endogenous housekeeping control gene. Each PCR reaction was performed in a ten μL mixture containing 0.25 U GoTaq G2 Hot Start Polymerase (Promega, Madison, WI, USA), 1 × GoTaq G2 Buffer, 1.5 mM MgCl2 (Invitrogen, Waltham, MA, USA), 200 nM of each primer (forward and reverse), 200 μM dNTP mixture containing the four deoxyribonucleotides (dATP, dCTP, dTTP, and dGTP), and ten ng of cDNA template. Appropriate negative controls and genomic DNA positive controls were incorporated into each experiment. Amplification thermal program included an initial denaturation step of 95 °C for 3 min and 40 cycles of 95 °C for 15 s, 60 °C for 15 s and 72 °C for 15 s using the ProFlexTM PCR System Thermal Cycler (Applied Biosystems, Waltham, MA, USA). Subsequently, the total amount of PCR product was separated by electrophoresis at 110 V for 40 min in 1.8% agarose gel diluted in 1 × Tris-acetate EDTA buffer (w/v) and stained with 0.01% of SYBR Safe (Thermo Fisher, Waltham, MA, USA).

For real-time quantitative PCR, the reactions were carried out in triplicates in a reaction mixture containing 1 × GoTaq qPCR MasterMix (Promega Corporation, Madison, WI, USA), 300 nM CXR Reference Dye, a final concentration of 200 nM of each (forward and reverse) SYBR green-designed primers (Thermo Fisher Scientific, Waltham, MA, USA), and ten ng of cDNA template per reaction. Appropriate negative controls were added in each run. The relative expression of the genes of interest: ACE2 (forward: 5′-CGAAGCCGAAGACCTGTTCTA-3′; reverse: 5′-GGGCAAGTGTGGACTGTTCC-3′), MYH6 (forward: 5′-GCCCTTTGACATTCGCACTG-3′; reverse: 5′-GGTTTCAGCAATGACCTTGCC-3′), MYH7 (forward: 5′-TCACCAACAACCCCTACGATT-3′; reverse: 5′-CTCCTCAGCGTCATCAATGGA-3′) was normalized by human reference genes: Glyceraldehyde-3-phosphate Dehydrogenase (GAPDH; forward: 5′-GCCCTCAACGACCACTTTG-3′; reverse: 5′-CCACCACCCTGTTGCTGTAG-3′) and Hypoxanthine Phosphoribosyltransferase 1 (HPRT-1; forward 5′-CGTCGTGATTAGTGATGATGAACC-3′; reverse: 5′-AGAGGGCTACAATGTGATGGC-3′). The reactions were performed on a StepOnePlusTM Real-Time PCR System thermocycler (Applied Biosystems, Waltham, MA, USA). Thermal cycling program comprised of a denaturing step at 95 °C for 3 min, followed by 40 cycling stages at 95 °C for 15 s, 57 °C for 15 s, 72 °C for 15 s and melt curve stage 95 °C, 15 s; 60 °C, 1 min; 95 °C, 15 s. Data analysis was performed with LinRegPCR quantitative PCR data analysis program v. 2020.0, as previously described.

Immunofluorescence staining

SARS-CoV-2-infected and mock-treated hiPSC-CMs were fixed using 4% paraformaldehyde solution (Sigma-Aldrich, EUA, St. Louis, Missouri, USA) for 1 h and stored at 4 °C. Next, cells were washed with PBS and then incubated with permeabilization/blocking solution (0.3% Triton X-100 /1% bovine serum albumin + 3% normal goat serum) for 1 h. Cardiomyocytes were incubated with primary antibodies diluted in a blocking buffer solution at 4° overnight: anti-SARS-CoV-2 convalescent serum from a positive COVID-19 patient (1:1,000) and anti-cardiac troponin T (TNNT2) (1:500, MA5-12960; Invitrogen, St. Louis, Missouri, USA). Afterwards, cardiomyocytes were incubated with the secondary antibody diluted in a blocking buffer solution: goat anti-Human Alexa Fluor 647 (1:400; A-21445; Invitrogen, St. Louis, Missouri, USA) and goat anti-Mouse 594 (1:400; A-11032; Invitrogen, St. Louis, Missouri, USA) for 1 h. Actin filaments were stained with Alexa Fluor 568 phalloidin (1:10; A-12380; Life Technologies, Carlsbad, California, USA) for 1 h. Nuclei were stained with 300 nM 4′-6-diamino-2-phenylindole (DAPI) for 5 min and each well was mounted with two drops of 50% PBS-Glycerol. Images (at least ten fields per well) of hiPSC-CMs were acquired using Operetta® High-Content Imaging System (Perkin Elmer) with a 20 × long working distance (WD) objective lens. A Leica TCS-SP8 confocal microscope was used to acquire images of hiPSC-CMs immunostained for TNNT2 and F-actin with the 63 × objective (Fig. S4).

Neutral red uptake cell viability assay

Briefly, hiPSC-CMs were seeded in 96-well plates. After reaching 80–90% confluency, cells were exposed to concentrations of WIN ranging between 10 nM–10 μM for 72 h. Next, the medium was replaced, cells were washed with PBS 1× and 200 μL of neutral red dye diluted in the hiPSC-CMs medium was added to each well at a final concentration of 0.05 mg/mL. After 3 h of incubation at 37 °C, neutral red dye was removed, and the cells were washed again. Then, 100 μL of the neutral red desorb solution was added (1% acetic acid-49% ethanol) to the wells, followed by 20 min in orbital shaking. Absorbance at 540 nm was measured with a Tecan Infinite® 200 PRO (Life Sciences, Switzerland) spectrophotometer.

Western blotting

Twenty-four hours after treatment with WIN of hiPSC-CMs in 24-well plates, 100 µL of sample buffer without bromophenol blue (62.5 mM Tris-HCl, pH 6.8, containing 10% glycerol, 2% SDS, and 5% two-mercaptoethanol) was added in each well, and a cell scraper was used to help lyse the cells. Cell extracts were transferred to an Eppendorf tube, boiled at 95 °C for 5 min, and centrifuged at 4 °C 16,000 ×g for 15 min to collect the supernatant. Protein content was estimated using the Bio-Rad Protein Assay (# 5000006; Biorad, Hercules, CA, USA). Next, bromophenol blue (0.02%) was added, and extracted samples (40 µg/lane) were separated by an 8% SDS polyacrylamide gel electrophoresis and transferred to polyvinylidene difluoride (PVDF) membranes. The membranes were blocked in 5% non-fat milk in Tris Buffered Saline with 0.1% Tween-20 (TBS-T) for 1 h at room temperature. Then, membranes were incubated overnight at 4 °C with primary antibodies anti-ACE2 (1: 1,000; MA5-32307; Thermo Fisher, Waltham, MA, USA), anti-CB1 (1:300; CSB-PA007048, Cusabio), and anti-ACTIN (1:2,000; MAB1501; Millipore, Burlington, MA, USA), diluted in TBS-T with 5% non-fat milk. Membranes were washed and incubated with peroxidase-conjugated antibodies IgG (H + L), HRP-conjugate: goat anti-mouse (1: 10,000, G21040; Molecular Probes, Eugene, OR, USA), goat anti-rabbit (1: 10,000, G21234; Molecular Probes, Eugene, OR, USA), and rabbit anti-goat (1: 2,000, 61–1620; Invitrogen, Waltham, MA, USA) for 2 h at room temperature. The signals were developed using an ECL Prime Western Blotting System (# GERPN2232; Sigma, St. Louis, Missouri, USA) for 5 min, and chemiluminescence was detected with an Odyssey-FC Imaging System® (LI-COR Biosciences, EUA). After CB1 or CB2 detection a stripping protocol was used on the membranes for further detection of actin. Membranes were incubated with a stripping buffer (pH 2.2, 200 mM glycine, 0.1% SDS, and 1% Tween-20) for three cycles of 10 min. Next, the buffer was discarded, and the membranes were washed three times with PBS and three times for 5 min with 0.1% TBS-T. Then, membranes were blocked again and proceeded with the above-described steps.

Statistics

Statistical analyses were performed using GraphPadPrism software version 8.0 (GraphPad, EUA). Results were expressed as the mean and standard error of the mean (SEM). For comparisons between two experimental groups, unpaired two-tailed Student’s t-test or Mann-Whitney U test was used, whereas two-way analysis of variance (ANOVA) or Kruskal–Wallis test followed by Tukey’s test was used for comparisons between three or more groups. A p-value smaller than 0.05 was accepted as statistically significant.

Ethics statement

Approved by the Research Ethics Committee of D’Or Institute of Research and Education (IDOR) 39474020.8.0000.5249.

Results

Human cardiomyocytes express cannabinoid receptor 1 but WIN does not modulate ACE2 expression

As a first step to investigate the influence of cannabinoid receptors in SARS-CoV-2 infection of human cardiomyocytes, we checked whether hiPSC-CMs expressed CB1 and CB2 receptors. We found that hiPSC-CMs express only CB1 receptor mRNA (Fig. S1A), which was confirmed by CB1 protein expression (Fig. S1B). The banding pattern observed in the hiPSC-CMs was similar to the mouse hippocampus sample (positive control) and consistent with what was observed in samples from other CNS regions (Medina-Vera et al., 2020).

WIN is an agonist at the CB1 and CB2 receptors with a higher affinity to them than other cannabinoids (Acheson et al., 2011; Sachdev et al., 2019), including THC (Felder et al., 1995) and therefore it is an useful pharmacological tool to study cannabinoid receptor activation. Beforehand, we tested multiple WIN concentrations in readouts of cellular toxicity and permanent cardiac hypertrophy. We found that WIN did not reduce cell viability in concentrations up to one µM (Fig. S3A). Also, compared with control, one µM WIN did not increase MYH6 and MYH7 mRNA levels (Fig. S3B), genes that, when upregulated, may indicate cardiac hypertrophy in vitro (Wenzel, 1967; Rahmatollahi et al., 2016; Albakri, 2019). Therefore, we chose one µM WIN as the usage concentration for further assays since it caused neither cell death nor changes in gene expression related to hypertrophy.

After confirming that hiPSC-CMs express CB1 and ACE2 (Salerno et al., 2021), we asked whether WIN modulates ACE2 expression and, subsequently, influences SARS-CoV-2 infection within hiPSC-CMs. The cells were pretreated with one µM WIN for 24 h and analyzed for both mRNA and protein levels of ACE2. We observed that WIN-treated (1.15 ± 0.07 A.U.) and untreated (0.98 ± 0.11 A.U.) hiPSC-CMs had comparable levels of ACE2 mRNA whereas ACE2 protein levels in WIN treated cells were 1.16 ± 0.39, normalized to control (Figs. 1A, 1B and S2).

Figure 1 WIN does not modulate ACE2 in hiPSC-CMs.

(A) Relative mRNA expression levels of ACE2 in WIN-treated hiPSC-CMs expressed as fold change relative to untreated condition. (B) Quantification of western blots by densitometry normalized by actin expression. ACE2 mRNA and protein levels were comparable between WIN-treated and untreated hiPSC-CMs. Error bars represent standard errors of the means (SEMs) from three (A) and four (B) independent experiments (three or four cellular differentiations) from one cell line.

WIN does not influence SARS-CoV-2 infection and replication in hiPSC-CMs

Next, we asked whether WIN could reduce hiPSC-CMs SARS-CoV-2 infection by mechanisms other than ACE2 modulation. For this, cells were pretreated with one µM WIN for 24 h and infected with SARS-CoV-2 at a multiplicity of infection (MOI) of 0.1 for 1 h, and the PFU analyzed 48 h later. In this study, we defined the use of MOI 0.1 for all experiments because this MOI had already been successfully used to infect hiPSC-CMs with SARS-CoV-2 (Sharma et al., 2020). Additionally, MOIs above 0.1 may not be a clinically plausible viral load found in vivo. Forty-eight hours after infection, we quantified convalescent serum (CS)-immunostaining and, as expected, we found that cells in the MOCK group had no CS immunoreactivity. Among the SARS-CoV-2-infected cells, those pretreated with WIN had a comparable percentage of infected cells (WIN SARS-CoV-2; 30 ± 15%) with those untreated (SARS-CoV-2; 26 ± 12%) (Figs. 2A and 2B).

Figure 2 WIN does not reduce SARS-CoV-2 infection and replication in hiPSC-CMs.

(A) Representative micrographs of MOCK and SARS-CoV-2-infected hiPSC-CM pretreated or not with one μM WIN for 24 h. hiPSC-CM were immunostained with SARS-CoV-2 convalescent serum (CS) (red) and counterstained with DAPI (blue) at 48 h post-infection. Scale bar: 50 μm. (B) Percentage of CS positive cells. CS immunoreactivity was comparable between treated and untreated hiPSC-CM. (C) Viral titer quantification by plaque forming units assay using the supernatants of the SARS-CoV-2 infected hiPSC-CMs. Viral titer was comparable between treated and untreated hiPSC-CM. Error bars represent standard errors of the means (SEM) from three independent experiments (three cellular differentiations and three independent infections) from one iPSC line.

Since viral infection and replication are correlated but orchestrated by different mechanisms, we asked whether WIN could decrease SARS-CoV-2 replication in hiPSC-CMs. We observed that despite a decrease in average viral titer when comparing SARS-CoV-2 WIN (6.99 × 105 ± 4.39 × 105 PFU/mL) with SARS-CoV-2 (2.18 × 106 ± 9.96 × 105 PFU/mL), the difference was not statistically significant (Fig. 2C).

WIN reduces the secretion of inflammatory cytokines in SARS-CoV-2-infected hiPSC-CMs

The “cytokine storm” is a hallmark of severe COVID-19 cases and cannabinoids have well-known anti-inflammatory properties. We asked whether WIN could reduce the release of the inflammatory cytokines IL-6, IL-8, TNF-α by hiPSC-CMs in vitro. Cells were pretreated with one µM WIN for 24 h, infected for 1 h, and incubated further for 24, 48, and 72 h. Then, the media were harvested at each time point for analysis. We found that cells infected with SARS-CoV-2 released higher levels of cytokines when compared with MOCK, with the exception of IL-8 at 24 and IL-6 at 72 h post-infection (Figs. 3A, 3B and 3C). Most importantly, in all conditions that significantly augmented the release of these pro-inflammatory cytokines, WIN was able to prevent this increase (Figs 3A, 3B and 3C). Of note, whereas the basal amount of cytokines tended to increase during culture time as they accumulated without media changes, WIN did not significantly affect this basal release by comparing MOCK and MOCK WIN groups.

Figure 3 WIN reduces inflammatory markers and viral toxicity in SARS-CoV-2-infected hiPSC-CMs.

Levels of IL-6 (A), IL-8 (B), TNF-α (C), and (D) release of lactate dehydrogenase (LDH) from MOCK and SARS-CoV-2-infected hiPSC-CM, treated or not with one μM WIN for 24 h, were analyzed at 24-, 48-and 72-h post-infection (h.p.i.). Cytokine levels were higher in SARS-CoV-2 compared with control (MOCK), and lower in SARS-CoV-2 WIN when compared with SARS-CoV-2. LDH release-absorbance levels relative to MOCK were higher in SARS-CoV-2 compared with SARS-CoV-2 WIN. Data represent means and standard errors of the means (SEM) from three independent experiments from one cell line. *p < 0.05, **p < 0.01, ***p < 0.001, ****p < 0.0001.

WIN reduces cell death in SARS-CoV-2-infected hiPSC-CMs

It has been previously reported that SARS-CoV-2 infection causes apoptosis in hiPSC-CMs (Perez-Bermejo et al., 2021). As cannabinoids can be protective in some tissues, we investigated whether WIN would protect hiPSC-CMs from cell death. Cells were pretreated with one µM WIN for 24 h, infected for 1 h, and cultivated for additional 24, 48, and 72 h and LDH was measured in the media at these different time points. Forty-eight and 72 h after the infection with SARS-CoV-2 without WIN, the release of LDH increased 463% and 174%, respectively, in hiPSC-CMs. On the other hand, hiPSC-CMs infected with SARS-CoV-2 and exposed to WIN had significantly lower increments of 72% and 40%, respectively (Fig. 3D).

Discussion

Cannabinoids have been proposed as potential treatment and prevention of COVID-19, due to their antiviral, cytoprotective and anti-inflammatory properties (Marchalant, Rosi & Wenk, 2007; Rossi et al., 2020; Anil et al., 2021). In this study, we showed that the synthetic CB1/CB2 agonist WIN reduced cell damage in SARS-CoV-2-infected hiPSC-CMs. Additionally, even though cardiomyocytes are not known for evoking robust inflammatory responses, WIN reduced the release of cytokines by these cells following SARS-CoV-2 infection. To our knowledge, this is the first study showing anti-inflammatory and protective properties of a cannabinoid agonist in hiPSC-CMs infected with SARS-CoV-2.

We hypothesized that WIN reduces the levels of ACE2 in hiPSC-CMs, consequently abrogating SARS-CoV-2 infection and viral load in these cells. However, despite a tendency towards increased ACE2 expression in hiPSC-CMs, it was not modulated by WIN in the conditions studied here. ACE2 is downregulated in SARS-CoV-2 infected tissues (Yan et al., 2020; Gheblawi et al., 2020), which is harmful to the heart since ACE2 has a protective role in the cardiovascular system (Huentelman et al., 2005; Zhong et al., 2010). Studies have shown that agonists of cannabinoid receptors, including WIN, cause vasodilation through the activation of CB1 receptors, and are capable of modulating vasoactive ligands (Sainz-Cort & Heeroma, 2020; Miklós et al., 2021). One possibility for the tendency towards an increase in the levels of ACE2 in WIN-treated hiPSC-CMs is that this cannabinoid agonist could exert a protective role by preventing receptor downregulation. Although it has been previously shown (Wang et al., 2020) that CBD-rich extracts reduced ACE2 mRNA and protein levels in some epithelia in vitro following TNF-α insult, this modulation had not been investigated in SARS-CoV-2 infected cardiomyocytes until now.

Despite evidence of cannabinoid receptors expression in murine embryonic stem cells (Jiang et al., 2007) and human cardiomyocytes (Mukhopadhyay et al., 2010), to our knowledge, this is the first description of the expression of CB1 receptor in hiPSC-CMs. The modulation of cannabinoid receptors in cardiomyocytes has also not been explored yet. Our results showed that WIN did not reduce the infection rate or the viral titer in hiPSC-CMs in the conditions studied here. Several studies have examined the effect of cannabinoids on viral infections, especially regarding the role of CB1 and CB2 receptor activation (Reiss, 2010). The CB2 receptor agonist JWH-133 reduced CXCR4-tropic HIV-1 infection of primary CD4+ T cells, whereas the CB1 receptor agonist arachidonoyl-29-chloroethylamide had no effect. In another study with HIV-1-infected primary human monocytes, agonists of CB2 receptors limited viral replication (Ramirez et al., 2013). There is still no consensus on the antiviral mechanisms of cannabinoids, however, it is well-known that the selective activation of the CB2 receptor plays a crucial role in the course of viral infection (Rossi et al., 2020). The fact that hiPSC-CMs do not express the CB2 receptor may explain WIN’s ineffectiveness in reducing SARS-CoV-2 infection and replication in these cells. Additionally, to date, cannabinoid treatment along with SARS-CoV-2 infection had not been investigated in this cellular model. It is likely that viral infection mechanisms through CB1 and CB2 receptors might vary depending on virus and cell type (Reiss, 2010; Tahamtan et al., 2016).

Although immune cells and cardiac fibroblasts are typically the major players in cytokine production under stressed cardiac conditions (Zhong et al., 2010), cardiomyocytes are also a local source of proinflammatory cytokines (Yamauchi-Takihara et al., 1995; Ancey et al., 2002; Kleinbongard, Schulz & Heusch, 2011; Atefi et al., 2011; Bozzi et al., 2019). In this work, hiPSC-CMs released IL-6, IL-8, and TNF-α at baseline levels and SARS-CoV-2 infection increased all cytokines levels. It has been shown that infection of hiPSC-CMs by Trypanosoma cruzi, the Chagas’ disease pathogen, prompted these cells to produce proinflammatory cytokines that caused autocrine cardiomyocyte dysfunction (Bozzi et al., 2019). Cardiac damage in COVID-19 patients can be attributable to hypoxemia due to respiratory dysfunction (Guo et al., 2020) but also to the “cytokine storm”, which is the uncontrolled systemic inflammatory response likely caused by an imbalance between regulatory and cytotoxic T cells (Meckiff et al., 2020). Even though the “cytokine storm” is one of the hallmarks of SARS-CoV-2 infection (Coperchini et al., 2020), one cannot rule out that cytokines locally released contribute to tissue damage, as seen, for example, in Trypanosoma cruzi cardiac infection (Bozzi et al., 2019). Here we found that WIN decreased the levels of IL-6, IL-8, and TNF-α released by SARS-CoV-2-infected hiPSC-CMs. An in vitro study of cortical astrocytes treated with Amyloid-β1–42, which is a neurotoxic protein, showed that WIN reduced TNF-α and IL-1β levels, while preventing cell death (Aguirre-Rueda et al., 2015). In another study, WIN decreased the activity of peroxisome proliferator-activated receptor alpha and TNF-α levels in the heart tissue of mice with cardiac dysfunction (Rahmatollahi et al., 2016), reinforcing its anti-inflammatory and protective properties in cardiac tissue.

THC and WIN are structurally different and accordingly have different efficacies towards activation of cannabinoid signaling pathways (Soethoudt et al., 2017). However, they are both mutual CB1 and CB2 receptor agonists (Compton et al., 1992) and can produce similar pharmacological effects, depending on the assay (Fan et al., 1994). THC presented a protective role against hypoxia in neonatal murine cardiomyocytes by reducing the levels of LDH (Shmist et al., 2006). Since neonatal murine cardiomyocytes expressed CB2, but not CB1, the authors suggest that cardioprotection provided by THC occurs via the CB2 receptor. Herein, we were able to show that hiPSC-CMs expressed CB1 but not CB2. In human cardiomyocytes, WIN decreased the release of LDH release and this effect could be mediated by CB1, to which WIN has high affinity. Other receptor candidates, such as transient receptor potential vanilloid (TRPV) channels (Freichel et al., 2017) can not be discarded. Nonetheless, the modulation of TRPV channels by WIN occurs at ten µM or higher (Jeske et al., 2006; Koch et al., 2011), which are, at least, ten times above the concentration used in our study.

Conclusion

This study showed that pretreatment with a cannabinoid receptor agonist reduced cytotoxicity and the release of proinflammatory cytokines by human cardiomyocytes infected with SARS-CoV-2. These results suggest that the therapeutic potential of cannabinoids in protecting the heart against SARS-CoV-2 infection should be further explored, in particular regarding selective action on the CB1 receptor.

Supplemental Information

Supplemental Information 1 CB1 and CB2 receptors expression.

(A) The figure shows agarose gel of end-point PCR products for cDNA from hiPSC-CMs. One single specific band was detected for CB1 receptor (125 bp), while no such band was detected for CB2 receptor (172 bp). GAPDH (352 bp) was used as an endogenous control to confirm efficiency of the amplification reaction and quality of cDNA template. Data from two independent experiments. (B) Western blot detection of CB1 receptor protein levels in hiPSC-CMs. The arrows indicate the specific bands, corresponding to the molecular weight predicted to CB1 receptor. hiPSC-CMs samples were taken from different passages and the positive control (CT) was from tissue homogenates of adult Black C57/BL6 mouse hippocampus.

Click here for additional data file.

Supplemental Information 2 Full lenght uncropped blot from Fig. 1.

After transference the membrane was cut, the upper half used for ACE2 detection, and the bottom half used for actin detection. Images of the membranes used for detection with the ladder (Amersham ECL Rainbow Marker-Full range) are shown on the left. Full-length gels for ACE2 and actin with contrast adjusted to allow visualization of the membrane are shown on the right. Although the whole gel is shown here the lanes used for representative image in Fig. 1 are highlighted in a box. Other lanes have samples that are not related to the experiment in this manuscript.

Click here for additional data file.

Supplemental Information 3 Cell viability assay and quantitative real‑time PCR for genes correlated to hypertrophy.

(A) Neutral red uptake assay from hiPSC-CMs treated with increasing concentrations of WIN for 72 h. The highest non-cytotoxic concentration was one μM. (B). qPCR for MYH6 and MYH7 genes from hiPSC-CMs treated with one μM WIN for 24 h. The MYH6 and MYH7 levels showed no significant differences between WIN-treated and untreated cells.

Click here for additional data file.

Supplemental Information 4 Micrographs of hiPSC-CMs immunostained for cTnT.

hiPSC-CMs were immunostained for TNNT2 (red), filamentous actin (F-actin) (green) by phalloidin staining and counterstained with DAPI (blue); 63 × magnification; Scale bar: 50 μm.

Click here for additional data file.

Supplemental Information 5 Raw data of all figures.

Click here for additional data file.

Additional Information and Declarations

Competing Interests

Author Contributions

Human Ethics

Data Availability

Diogo Biagi, Estela M. Cruvinel and Rafael Dariolli are employed by Pluricell Biotech.

Luiz Guilherme H. S. Aragão conceived and designed the experiments, performed the experiments, analyzed the data, prepared figures and/or tables, authored or reviewed drafts of the paper, and approved the final draft.

Júlia T. Oliveira conceived and designed the experiments, performed the experiments, analyzed the data, prepared figures and/or tables, authored or reviewed drafts of the paper, and approved the final draft.

Jairo R. Temerozo conceived and designed the experiments, performed the experiments, analyzed the data, authored or reviewed drafts of the paper, and approved the final draft.

Mayara A. Mendes performed the experiments, analyzed the data, prepared figures and/or tables, authored or reviewed drafts of the paper, and approved the final draft.

José Alexandre Salerno performed the experiments, analyzed the data, authored or reviewed drafts of the paper, and approved the final draft.

Carolina S. G. Pedrosa performed the experiments, authored or reviewed drafts of the paper, and approved the final draft.

Teresa Puig-Pijuan performed the experiments, analyzed the data, authored or reviewed drafts of the paper, and approved the final draft.

Carla P. Veríssimo performed the experiments, prepared figures and/or tables, authored or reviewed drafts of the paper, and approved the final draft.

Isis M. Ornelas performed the experiments, analyzed the data, prepared figures and/or tables, authored or reviewed drafts of the paper, and approved the final draft.

Thayana Torquato performed the experiments, authored or reviewed drafts of the paper, and approved the final draft.

Gabriela Vitória performed the experiments, authored or reviewed drafts of the paper, and approved the final draft.

Carolina Q. Sacramento performed the experiments, authored or reviewed drafts of the paper, and approved the final draft.

Natalia Fintelman–Rodrigues performed the experiments, authored or reviewed drafts of the paper, and approved the final draft.

Suelen da Silva Gomes Dias performed the experiments, authored or reviewed drafts of the paper, and approved the final draft.

Vinicius Cardoso Soares performed the experiments, authored or reviewed drafts of the paper, and approved the final draft.

Letícia R. Q. Souza performed the experiments, authored or reviewed drafts of the paper, and approved the final draft.

Karina Karmirian performed the experiments, prepared figures and/or tables, authored or reviewed drafts of the paper, and approved the final draft.

Livia Goto–Silva performed the experiments, authored or reviewed drafts of the paper, and approved the final draft.

Diogo Biagi performed the experiments, authored or reviewed drafts of the paper, and approved the final draft.

Estela M. Cruvinel performed the experiments, authored or reviewed drafts of the paper, and approved the final draft.

Rafael Dariolli performed the experiments, authored or reviewed drafts of the paper, and approved the final draft.

Daniel R. Furtado analyzed the data, authored or reviewed drafts of the paper, and approved the final draft.

Patrícia T. Bozza analyzed the data, authored or reviewed drafts of the paper, and approved the final draft.

Helena L. Borges analyzed the data, authored or reviewed drafts of the paper, and approved the final draft.

Thiago M. L. Souza analyzed the data, authored or reviewed drafts of the paper, and approved the final draft.

Marília Zaluar P. Guimarães conceived and designed the experiments, analyzed the data, prepared figures and/or tables, authored or reviewed drafts of the paper, and approved the final draft.

Stevens K. Rehen conceived and designed the experiments, analyzed the data, authored or reviewed drafts of the paper, and approved the final draft.

The following information was supplied relating to ethical approvals (i.e., approving body and any reference numbers):

This work was approved by the Ethics committee of D’Or Institute of Research and Education (IDOR) (39474020.8.0000.5249).

The following information was supplied regarding data availability:

The raw data is available as a Supplementary File.

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
