# Peer review of "WIN 55,212-2 shows anti-inflammatory and survival properties in human iPSC-derived cardiomyocytes infected with SARS-CoV-2"

_PeerJ, doi:10.7717/peerj.12262_

## Round 0.1 · original submission · Minor Revisions

Both Reviewers thought that that this work is potentially important and interesting. Both are cannabinoid receptor experts, and picked up on the slightly loose terminology/definitions around cannabinoids in the manuscript - for example the characterization of WIN55212 as a "full agonist" (high efficacy would be better) with a "higher affinity" than other cannabinoids - true for some but not all synthetic cannabinoids, needs attention. There a number of other issues the reviewers raise that need attention in a revised manuscript - either revision or explanation.

However, the major weakness in the study is the failure of the authors to test the actual mechanism of action of WIN55212. They do not examine the involvement of CB receptors through use of antagonists, nor do they test the CB receptor antagonist enantiomer of WIN - WIN55212-3. While it is reasonably likely that WIN55212 is acting through cannabinoid receptors, unless you test it, you don't know. The authors provide evidence that the cells they are studying express CB receptors, but without an antagonist, the involvement of CB receptors is simply a guess. The authors do not claim that WIN55212 is acting through CB receptors in the Abstract, but somewhere in the manuscript, there needs to be an explicit statement that the involvement of CB receptors in these effects need to be tested - or indeed the authors should actually test it, this would make the paper a much stronger and potentially impactful piece of work.

Reviewer 1 ·

Basic reporting

The manuscript is well written and structured, but the authors would be appreciated if they clarified the following:


1. Suggested title: WIN55,212-2 shows anti-inflammatory and survival properties in human…


2. Line 85 : …elevated levels of creatine kinase and lactate dehydrogenase activity (LDH), which are injury biomarkers... To which organ/system?

3. Line 88-90: The citations do not appear to support the claim of correlation between high expression of ACE2 and patients susceptibility to severe COVID 19.

4. Line 106-108: Studies have been investigating… but only one example is given. Citations needed.
5. Line 110-113: High affinity of WIN to CB receptors does not necessarily imply high functional activity or efficacy. Author should consider citation of functional data of WIN

6. Line 113-122: The author has given good examples to support anti-inflammation by WIN, but these activities were not CB-mediated. Author should include studies such as PMID: 10734163, which are examples of CB mediated anti-inflammation. A statement highlighting that the anti-inflammatory activity of WIN may or may not be CB mediated will improve this section.
7. Line 124-125: Citation needed.
8. Line 170: Full meaning of MOI at first use.
9. Line 275-276: Same as line 110. Reference to functional data will be more suitable.

Experimental design

10. Line 277: There is significant evidence that CB2 antibodies have poor selectivity and/or specificity (Atwood, Straiker, & Mackie (2012), Cécyre et al. (2013), Marchalant et al. (2014). I) Did you use a robust negative control (e.g. undifferentiated iPSC) in your western blot? Ii)How many biological replicates did you undertake? Iii) Did you validate the CB2 antibody? V) Could you please provide densitometry for CB receptor expression? Iv) Please include a figure legend in your supplementary data.

11. Line 280: I may be mistaken, but Marchalant, Rosi & Wenk, 2007 used immunohistochemistry (IHC) to study CB1 expression. How do you compare the banding pattern of WB to IHC? Secondly, please cite the original article; Atwood & Mackie, 2010 is a review.

12. There are several pro-inflammatory cytokines. What was the rationale for choosing these three?

Validity of the findings

The studies are robust except for the data which support the expression of CB receptors. The relevant clarifications have been raise in the experimental section above.

Additional comments

13. Line 328: …which is a hallmark for damage. Damage to what? Cell damage?

14. Line 334: Discussion can be improved by elaborating upon the CB1 and CB2 expression on hiPSC-CMs and comparing to other literature. There is evidence that both receptors are expressed in murine embryonic stem cells (Jiang et al. 2007), but perhaps there is little information on hiPSC-CMs

15. Line 368-369: If your data shows CB receptor expression. How does the absence of CB2 receptors explain the inactivity of WIN?

16. Line 393-400: THC and WIN are CB1 and CB2 agonists, but their efficacy is significantly different (Soethoudt et al. 2017). So, they will not necessarily produce a similar effect through the cannabinoid receptors.

17. Line 395-400: CB1 and CB2 receptors mediate cardioprotective properties (Liao et al. 2013; Batkai et al. 2007; Dueer et al. 2014; Netherland et al. 2010). Therefore cardioprotection by THC in neonatal murine cardiomyocytes could have been mediated through CB2 because CB1 receptors were not expressed. Although the authors subsequently mention CB1, I’ll suggest the inference in lines 399-400 should mention both receptors instead of highlighting CB2 receptors only.

·

Basic reporting

I commend the authors for undertaking this interesting study on investigating the anti-inflammatory potential of synthetic cannabinoid, WIN55212-2, in human iPSC-derived cardiomyocytes (hiPSC-CMs) infected with SARS-CoV-2. The authors of the present manuscript addressed this issue by examining the in vitro effect of WIN on modulating the expression of ACE2; cellular staining and confocal microscopy to study the effect of WIN on SARS-CoV-2 infection and replication; and the effect of WIN in modulating the release of several pro-inflammatory cytokines - IL-6, IL-8, and TNF-α associated with the disease.

In general, the manuscript should have been read carefully before submission. The paper is generally well written and structured. However, in my opinion, the paper has some shortcomings in regard to sample size (n=3) and text, and I feel this unique dataset has not been utilized to its full extent. I have provided numerous remarks on texts, specific examples are given below.

Experimental design

The experimental design is appropriately described. The impetus for this series of studies is also defined in the Introduction. Although the sample size is low, I am unsure if enough sample size is considered for this study to be repeatable.

Validity of the findings

Based on what is provided, the data appears to be robust. That said, while the results are noteworthy, the study is somewhat descriptive, and additional investigation of the mechanism underlying the anti-inflammatory effect of WIN in cells infected with SARS-CoV-2 (involvement of CB1 or CB2) would greatly improve the rigor, reproducibility, and utility of this manuscript.

Additional comments

Line 101: well described effects of cannabinoids on cardiovascular disease - the authors should elaborate on this. I think that the cardiovascular effects of cannabis are not well known. In this sentence, it is not very clear whether the authors are talking about the therapeutic or adverse effects of cannabis consumption on cardiovascular events.

Line 103: The author should also mention the other phytocannabinoid - CBD (non-psychoactive component of the cannabis plant). It is mentioned later in the text but needs an introduction here.

Line 104-108: Please provide a reference for the following statement - studies investigating the potential of cannabinoids on SARS-Cov-2 infection.

Line 117-120: It is important to mention the composition of cannabis sativa arbel strain, that it contains CBD, THCV, CBG, and multiple terpenes, as the same study has looked at the differences in the anti-inflammatory effects of CBD by itself compared to F(CBD) strain in alveolar epithelial cells.

Line 149: Consider rewording as this is somewhat confusing to read.

Line 206: Please indicate the incubation time for WIN treatment for each experimental setup in the methods section.

Line 261 and following: Group size should be included in the data and statistical analysis section - three independent experiments, in your case? I am not sure if the sample size of 3-4 is statistically appropriate for estimating the P-value using unpaired-test or ANOVA.

Line 275: I would not say that WIN is a full agonist, there are other synthetic cannabinoids identified in the recreational market that activates the CB1 receptor with much higher potency and efficacy than that of WIN. Please refer to this paper https://doi.org/10.1111/bph.14829

Line 279: Fig S1, please label the band appropriately. It is not very clear what I am looking at.

Line 316 and following: The authors showed that WIN was able to reduce the inflammatory cytokines in cells infected with SARA-CoV-2. Was this effect mediated by the CB1/2 receptor - it could have been tested simply by pre-treating the cells with CB1 or CB2 antagonist.

Line 330-333: This sentence should be rewritten for clarity as the subjects and descriptors may become confusing to the reader.

Line 368: This statement is a bit contradictory to what has been shown in the results section - Fig S1 indicates that hiPSC-CMs used in this study express both CB1 and CB2 receptors, so how does the ineffectiveness of WIN in reducing SARS-CoV-2 infection attributed to CB2 receptor?

Line 400 and following: there is a discussion on the involvement of CB2 receptors regarding the reduced levels of LDH by WIN. Due to lack of relevant data I find this discussion highly speculative, please rewrite the sentence, or omission is recommended.

Figure 3, It seems like asterisks representing statistical significance are missing from Figure 3D (72h incubation, SARS-Cov-2 ± WIN).

Textual remarks:
- Abstract Line 57: replace “provoke” with “may cause”
- Abstract Line 68: need internal consistency for SARS-CoV-2
- Line 86: replace “which are injury biomarkers” by “a biomarker of cell/tissue injury”
- Line 103: replace “agent” with “ingredient”
- Line 121: need internal consistency for hyphenation (pre-treated or pretreated)
- Line 125: need reference to support the statement
- Line 138: solution instead of solutions
- Line 172: …with or without treatment…instead of with treatments or not
- Line 202: add respectively at the end of the sentence
- Line 208: need consistency for hour or h
- Line 211: ...overnight: anti-SARS-Cov-2 convalescent… add a colon, and remove the bracket
- Line 226: don’t need to expand it every time - WIN is fine
- Line 228-229: please provide the full form of an abbreviation when it is used first time in the paper, neutral red dye (NR) is an example
- Line 230: …cells were washed again…
- Line 235: after 24h treatment of hiPSC-CMs...with what?
- Line 241: …and extracted samples…
- Line 247: comma missing after bracket enclosing Sigma-Aldrich
- Line 283: replace “regarding” with “for”
- Line 328: …hallmark for cell/tissue damage…

---

## Round 0.2 · accepted · Accept

Thank you for so comprehensively responding to the comments of the Reviewers. The major overall limitation of the study is the reliance on an agonist only, but the investigators have been careful to temper their conclusions about possible sites of action for the cannabinoid WIN55212, and as it stands, this work will add to our information about the potential role of cannabinoids in viral-induced inflammation, and will not doubt stimulate further work.